# Learning to Learn with Feedback and Local Plasticity

**Jack Lindsey, Ashok Litwin-Kumar**
Columbia University, Department of Neuroscience
{j.lindsey, a.litwin-kumar}@columbia.edu

## Abstract

Interest in biologically inspired alternatives to backpropagation is driven by the desire to both advance connections between deep learning and neuroscience and address backpropagation's shortcomings on tasks such as online, continual learning. However, local synaptic learning rules like those employed by the brain have so far failed to match the performance of backpropagation in deep networks. In this study, we employ meta-learning to discover networks that learn using feedback connections and local, biologically inspired learning rules. Importantly, the feedback connections are not tied to the feedforward weights, avoiding biologically implausible weight transport. Our experiments show that meta-trained networks effectively use feedback connections to perform online credit assignment in multi-layer architectures. Surprisingly, this approach matches or exceeds a state-of-the-art gradient-based online meta-learning algorithm on regression and classification tasks, excelling in particular at continual learning. Analysis of the weight updates employed by these models reveals that they differ qualitatively from gradient descent in a way that reduces interference between updates. Our results support the view that biologically plausible learning mechanisms may not only match gradient descent-based learning, but also overcome its limitations.[1]

## 1 Introduction

Deep learning has achieved impressive success in solving complex tasks, and in some cases its learned representations have been shown to match those in the brain [14, 21, 23, 30, 36]. However, there is much debate over how well the backpropagation algorithm commonly used in deep learning resembles biological learning algorithms. Several key features of backpropagation do not obviously map onto biological implementations. One such feature is the requirement in backpropagation that feedback weights are exactly tied to feedforward weights, even as weights are updated with learning. Another is that backpropagation applies the derivatives of the forward-pass nonlinearities during the feedback pass, which would require that feedback pathways have knowledge of the state of feedforward pathways, likely at some time offset. The question of how credit assignment – the communication of appropriate learning signals to neurons upstream of behavioral outputs – can be implemented by biological circuits remains open. It also remains unclear whether feedback pathways in neural circuits are best thought of as implementing an approximation to backpropagation, or some other qualitatively different learning algorithm.

We propose a learning paradigm that aims to solve the credit assignment problem in more biologically plausible fashion. Our approach is as follows: (1) apply local plasticity rules in a neural network to update feedforward synaptic weights, (2) endow the network with feedback connections that propagate information about target outputs to upstream neurons in order to guide this plasticity, and (3) employ meta-learning to optimize feedback weights, feedforward weight initializations, and rates of synaptic plasticity. The purpose of the meta-learned feedback is to modulate upstream activity in

such a way that, when the local plasticity rule is applied, useful weight updates are performed. On a set of online regression and classification learning tasks, we find that meta-learned deep networks can successfully perform useful weight updates in non-readout layers. In fact, we find that feedback with local learning rules can match and sometimes outperform gradient descent as a within-lifetime learning algorithm.

## 2   Related Work

A body of research has investigated alternative algorithms to backpropagation that relax or eliminate the requirement of weight symmetry. One surprising set of results [20, 26] shows that random feedback weights are sufficient to support learning for simple tasks. Another family of methods, known as target propagation, uses a reconstruction loss to learn a feedback pathway that approximates the inverse of the feedforward pathway [5]. However, both of these approaches have been found to scale poorly to difficult tasks such as ImageNet classification [3].

A number of more recent methods have made additional progress on the weight symmetry problem by proposing more biologically realistic mechanisms to enforce approximate weight symmetry and thereby approximate gradient descent. Akrout et al. [1] and Kunin et al. [16] propose local circuit mechanisms that enforce approximate weight symmetry and approach backpropagation-level performance on ImageNet classification. Guerguiev et al. [8] pursue another approach to enforcing approximate weight symmetry, leveraging the observation that the discontinuity of spiking neurons allows for inference of their causal effects on downstream neurons. Lansdell et al. [19] propose an RL strategy that enables neurons to estimate gradient signals. In this work we explore a very different approach, eschewing any explicit or implicit constraints on the relationship between feedforward and feedback weights. We view our approach as complementary to those described above.

Standard deep learning approaches that use stochastic gradient descent for optimization notably fall short of human and animal learning in several key respects. In particular, such approaches have difficulty learning from few examples [18] and learning online from a stream of data with nonstationary statistics [28]. One approach to addressing these issues is meta-learning, in which a network's learning procedure itself is learned in an "outer loop" of optimization. The literature contains many particular instantiations of this general idea. One class of models explores the possibility of learning through the recurrent dynamics of the network [11, 34], and recent work has sought to connect this approach to neuroscience [35]. Another line of research, dating back to [32], allows learning to take place via synaptic weight updates in the network, and meta-optimizes the weight update procedure. Several authors [2, 24, 25, 31] have developed a particular version of this approach in which the form of synaptic updates is restricted to biologically motivated Hebbian learning rules. Another popular class of methods is gradient-based meta-learning [6], in which the network initialization is meta-optimized so that batch gradient descent will learn quickly from few examples of a new task. This method has recently been extended to the continual learning case, in which the "inner loop" optimization consists of many online gradient steps on a potentially nonstationary data distribution [12].

## 3   Contributions

Our work draws from the literature described above and introduces new features. In our model, inner loop learning takes place by synaptic weight updates according to a local, biologically motivated learning rule. Unlike in [2, 24, 25], our model explicitly tackles the credit assignment problem by enabling plasticity in early network layers introducing feedback weights that may be meta-optimized to provide rich error information. This flexibility to meta-learn how error information is propagated also differentiates our work from gradient-based meta-learning, bringing it closer to biological plausibility and, we find, providing performance benefits in some cases. Despite this flexibility, our approach constrains the learning procedure substantially compared to methods that learn through recurrent dynamics, as in [11, 34], or weight update functions parameterized by another neural network, as in [32]. We propose that our neurobiologically inspired constraints and incorporation of error-carrying feedback connections provide useful inductive biases for meta-learning while retaining sufficient flexibility.

Our empirical results show the promise of this approach, which proves surprisingly effective even given the rather primitive tools – direct, shallow, and fixed feedback pathways – provided to our meta-learner. These findings motivate extensions that scale and extend the method to apply to more complex and diverse problems. Furthermore, we make preliminary analyses of the learning strategies uncovered by our meta-learning algorithm. Our observations have implications both for biologists examining the role of feedback connections in the brain, and for machine learning practitioners in search of effective inductive biases to guide learning.

## 4  Method

See Figure 1 for a schematic comparing our framework to standard backpropagation and direct feedback alignment [26]. Our method consists of three main stages – feedforward processing, feedback updates, and weight updates. For convenience, we will henceforth refer to the method as FLP, for "feedback and local plasticity."

First, a multi-layer feedforward network, whose $i$th layer has forward weights $\mathbf{W}_i$, propagates an input $\mathbf{x}$ forward through its layers, produces an output $\hat{\mathbf{y}}$, and receives a target signal $\mathbf{y}$. Then $\mathbf{y}$, or the prediction error $\mathbf{y} - \hat{\mathbf{y}}$, is propagated through a set of feedback weights. We obtained better performance using prediction errors for the regression task, and using raw targets for the classification task (see Section 5 for task details). Our reported results reflect these choices, which yield better performance both for our method and the gradient-based baseline, and thus do not unfairly advantage one method over the other. In our experiments, separate pathways carry feedback information directly from the output to each layer, as in direct feedback alignment [26]. The feedback to the $i$th layer layer takes the form of a single linear transformation, parametrized by the matrix $\mathbf{B}_i$. These choices were made for simplicity, and more complex feedback architectures are an interesting topic for future study.

Subsequently, the activations of the neurons at each layer are updated in response to the feedback. The activations of layer $i$, which are $\mathbf{x}_i$ during the feedforward pass, are updated to $\mathbf{x}_i \leftarrow (1 - \beta_i)\mathbf{x}_i + \beta_i \cdot \text{ReLU}(\mathbf{B}_i\mathbf{y} - \mathbf{b})$. Here, $\beta_i$ controls the strength of feedback relative to feedforward input, $\mathbf{B}_i$ is the matrix of feedback weights described above, and $\mathbf{b}$ is a bias term. The rectification ensures nonnegative activations following the feedback stage and introduces some nonlinearity in the feedback updates. Note that $\beta_i = 0$ corresponds to pure unsupervised Hebbian learning in layer $i$; thus, the $\beta_i$ parameter can be interpreted as interpolating between unsupervised and supervised learning.

The network then undergoes synaptic plasticity according to a local learning rule – local in the sense that a synaptic weight $w$ is updated based only on its existing value, the presynaptic activity $a$, and the postsynaptic activity $b$ resulting from feedback.[2] In our simulations we use Oja's learning rule: $w \leftarrow w + \alpha(ab - b^2 w)$, where $\alpha$ is a plasticity coefficient [27], a normalized modification of standard Hebbian learning that prevents diverging weights. We typically allow plasticity only in the final $N$ network layers, allowing the initial layers to serve as fixed feature extractors.

### 4.1  Meta-learning procedure

The description above specifies how a network in our model learns in its "lifetime." However, to create a network that effectively learns using the above procedure, we employ meta-learning. More specifically, for each of our benchmark tasks (Section 5) we simulate a lifetime consisting of an entire learning episode and a test input, evaluate the performance on the test input, backpropagate through the entire learning procedure (see [6, 12]), and repeat this process for many lifetimes. The meta-learned parameters are the initializations of $\mathbf{W}_i$, the feedback weights $\mathbf{B}_i$, as well as the plasticity

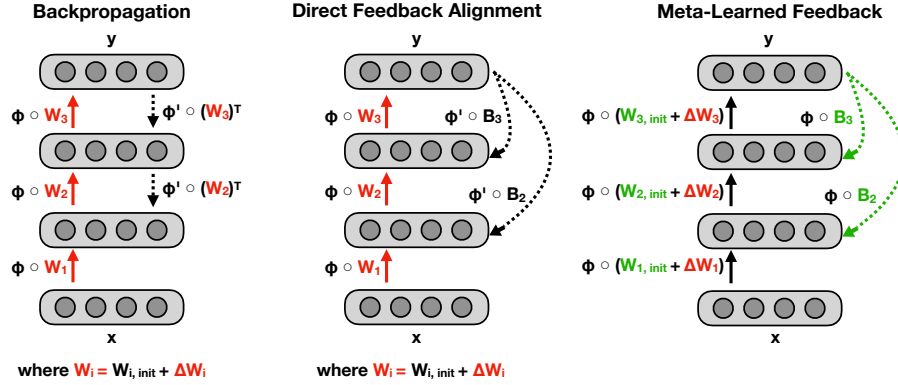

Figure 1: A comparison of backpropagation, direct feedback alignment [26], and the proposed method (FLP). $\mathbf{W}$ and $\mathbf{B}$ variables represent linear transformations, $\phi$ indicates the activation function, and $\circ$ denotes composition. Red quantities indicate plastic weights that change during a network's lifetime, while green quantities indicate meta-learned quantities optimized over many lifetimes. In backpropagation, learning signals propagate through a feedback pathway involving transposes of the feedforward weights and the derivative of the neuron activation function. Direct feedback alignment replaces the transpose matrices with random feedback pathways. In FLP, feedforward weights evolve according to Hebbian plasticity during a lifetime, while feedback pathways and initial feedforward weights are meta-optimized across many lifetimes. Additionally, error signals are injected into upstream layers directly, without any derivative computations.

coefficient $\alpha$ for each weight and the coefficient $\beta$ for each layer. We used the Adam optimizer [15] for meta-optimization. Additional implementation details can be found in Appendix B.

## 4.2 Universality

It can be shown that sufficiently wide and deep neural networks that employ the above learning procedure can approximate any learning algorithm. A learning algorithm, for our purposes, is a map from a set of training examples $\{(\mathbf{x}, \mathbf{y})_k\}$ and a test input $\mathbf{x}^\star$ to a predicted output $\hat{\mathbf{y}}^*$.

**Theorem.** Let $\theta$ refer to the feedforward weights of a network. For any learning rule $f_{\text{target}}(\{(\mathbf{x}, \mathbf{y})_k\}, \mathbf{x}^\star)$, there exists a deep ReLU network with associated feedforward function $\hat{f}(\cdot; \theta)$ and (potentially multilayer) feedback pathways as described above, such that $\hat{f}(\mathbf{x}^\star; \theta') \approx f_{\text{target}}(\{(\mathbf{x}, \mathbf{y})_k\}, \mathbf{x}^\star)$. Here $\theta' = \theta_k$, $\theta_0 = \theta$, and $\theta_{j+1} = \theta_j + \Delta_{\theta_j}(\mathbf{y}, \mathbf{x})$, where $\Delta_\theta(\mathbf{y}, \mathbf{x})$ is the weight update computed following feedback according to a local learning rule at each synapse, either Hebb's rule or Oja's rule.

**Proof.** See Appendix A for the complete proof. The proof borrows heavily from that of Finn et al. [7], but deviates from it in at least one major respect: in the online, continual learning case, the ability to choose feedback weights separately from the feedforward weights is essential to the proof construction. It should be noted that existence results of this kind have tenuous relationship with a method's practical utility; nevertheless we find this aspect of the proof suggestive of a key role – borne out in our experimental results – for decoupled feedforward and feedback weights in online learning.

## 5 Experiments

We build off the experimental protocol of Javed and White [12], evaluating our approach on the same regression and classification tasks, all of which require online learning. These tasks are themselves adaptations of those used in Finn et al. [6] to the online setting. We explore both online i.i.d. (data sampled randomly) and online continual (data from different distributions presented sequentially) learning. Also following [12], we use a nine-layer fully connected network for regression tasks, and a network with six convolutional layers + two fully connected layers for classification tasks. More details are provided in Appendix B.

Table 1: Regression Results (Mean squared error)

| Method | i.i.d. learning | Continual learning |
|---|---|---|
| Feature Reuse (1) | 0.050 (7e-3) | 0.035 (3e-3) |
| FLP (2) | 0.00093 (3e-5) | **0.0016 (6e-5)** |
| FLP (3) | **0.00051 (8e-5)** | 0.0068 (2e-3) |
| Gradient-based (3) | **0.00057 (2e-5)** | 0.069 (0.02) |
| Original OML (3) | 0.072 | 0.40 |

Table 2: Classification Results (% Error)

| Method | Omniglot | Omniglot (continual) | Mini-ImageNet |
|---|---|---|---|
| Feature Reuse (1) | 4.6 (0.1) | 5.0 (0.1) | 48.2 (0.1) |
| FLP (2) | **3.0 (0.1)** | **2.7 (0.1)** | **42.5 (0.6)** |
| Gradient-based (2) | **3.2 (0.1)** | 3.4 (0.1) | **42.5 (0.3)** |
| Original OML (2) | 3.5 | 7.0 | 52.0 |

*(Note: Parentheses in the "Method" column indicate the number of plastic layers in the network.)*

**Incremental Sine Waves:** The regression problem is as follows: in each training episode, ten sine functions $f_n(x)$, $n = 1 \ldots 10$, are sampled randomly, each parameterized by an amplitude in $[0.1, 5]$ and phase in $[0, \pi]$. The input $\tilde{x}$ contains both the function input $x$ and the index $n$ of the function ("sub-task") being used. The network must output $y = f_n(x)$. In each episode, 400 size-32 batches of $(\tilde{x}, y)$ pairs are presented, sampled equally from the ten sinusoids. In the i.i.d. version of the task, $(\tilde{x}, y)$ examples are presented in random order. In the continual learning variant, all examples from the first sinusoid are presented, then all from the second, and so on. We emphasize that in both variants, each size-32 batch of data is presented exactly once, without repetition, making the problem "online." At the end of an episode, the network is tasked with outputting $y$ for a new $\tilde{x}$. Evaluation occurs on new episodes with sine functions not used in meta-training. Meta-training is performed for 20,000 episodes.

**Online, few-shot classification:** We consider the Omniglot [17] and Mini-Imagenet [33] datasets. In each case, the dataset is split into meta-training and meta-testing classes. During an episode, $k$ examples from each of $N$ classes are presented. In the i.i.d. version of the task, they are presented in random order, while in the continual learning version, all $k$ examples from one class are presented before proceeding to the next. The model is tested on unseen examples from the classes in the episode. We evaluate performance for $k = 5$, $N = 5$. In the feedback phase, output activations are clamped to their target values, but feedback weights to earlier layers are meta-learned. Evaluation episodes use classes never seen in meta-training. Meta-training is performed for 40,000 episodes. We emphasize that in each episode, data is presented *one example at a time*, and each example is seen exactly once, distinguishing this task from typical $N$-way, $k$-shot classification benchmarks.

**Experimental Protocol:** We evaluate our method in two ways: (1) To assess our method's ability to enable useful deep credit assignment, we meta-train and test variants of the network with different numbers of plastic layers. We include as an important control the case in which only the output weights are plastic and thus no feedback is involved in learning. Following [29], we refer to this as the "feature reuse" regime, as such networks are constrained to fit readouts on top of a fixed feature extractor within each lifetime. (2) We compare our method's performance to a gradient-based meta-learner based on OML [12]) with the same architecture. We matched the architecture and hyperparameter optimization procedures of the two methods to enable fair comparison – this resulted in our baseline exceeding the performance of the unmodified OML algorithm (for which we also report results).

## 6   Performance Results

The discussion below refers to numerical results in Table 1 and Table 2. Listed figures reflect averages over two (for regression tasks) or five (for classification tasks) independently meta-trained networks, with standard error indicated in parentheses.

### 6.1 Meta-learned feedback is useful for learning

The objective of our framework is to enable deep credit assignment – namely, useful weight updates in non-readout layers – while avoiding biologically unrealistic model properties. To assess our method's ability to enable deep credit assignment, we meta-train and test variants of the network with different numbers of plastic layers, including the "feature reuse" regime where only output weights are plastic [29]. One of our central results is that, in all tasks we consider, enabling plasticity in non-readout layers improves performance (Tables 1 and 2), indicating that credit assignment is performed successfully.

### 6.2 FLP networks match gradient-based learners

These results demonstrate that meta-learning can uncover feedback weights that aid learning, without biologically implausible weight symmetry or nonlocality. Next we assess the significance of this improvement, relative to what can be achieved without biological constraints. For comparison, we consider a baseline adapted from OML [12], a state-of-the-art gradient-based online meta-learning model. We augmented the OML optimization procedure to match that of our model, such that the only difference is OML's use of gradient updates in its inner learning loop. As mentioned above, our modifications only improved performance of the original OML (details in Appendix B). We refer to this as our "gradient-based" baseline. In all cases, FLP matches the performance of the gradient-based baseline, indicating that meta-learned feedback weights can provide learning signals as useful as gradients, despite these weights being fixed rather than tied to feedforward updates. The learning trajectories are similar for FLP and gradient-based networks, with FLP networks appearing to learn more quickly on the Omniglot task (see Appendix C).

### 6.3 FLP networks outperform gradient-based learners on continual learning tasks

We next asked whether our method provides advantages *beyond* gradient-based learners. Motivated by the observation that gradient-based algorithms struggle on continual learning tasks, we experimented with a continual learning variant of the regression task. In this version, the network observes all data from one function before it encounters the next, and so on. Thus, the network is required to learn multiple functions sequentially within its lifetime, without losing its ability to generate previous functions. We found that on this task, our method yielded significantly better performance than the gradient-based baseline. Indeed, in the regression tasks, the gradient-based baseline did not outperform the feature reuse control, while FLP outperformed it substantially. This result suggests that our biologically motivated approach can not only match the performance of gradient-based algorithms, but in fact exceed them in difficult continual learning problems.

We also experimented with the continual learning variant of the Omniglot task, in which all examples of a given class are presented sequentially, followed by the examples of the next class. We found that FLP modestly outperformed the gradient-based baseline. We note that, due to computational constraints, the lengths of episodes in this task – 25 examples each – are much smaller than those of the continual learning regression task – 400 each – which may explain the more modest improvement. Future work may clarify the situations in which FLP is especially advantageous.

## 7 Analysis

The above results demonstrate the existence of learning algorithms that achieve high performance using weight updates that differ from those used by backpropagation. We next analyze properties of these algorithms to attain a better understanding of how they accomplish this.

### 7.1 Tradeoff between effective and adaptable feature extraction at initialization

We investigated whether networks with many plastic layers meta-learned a fundamentally different strategy than those with only a plastic readout limited to a "feature reuse" strategy. For convenience we will refer to the pre-readout component of a network as its "feature extractor". Networks constrained to a feature reuse strategy require a feature extractor that is meta-trained to compute generally useful features for the given family of tasks. Networks with plasticity in more layers, on the other hand, may adopt different strategies that involve adjustment of the feature extractor within a lifetime.

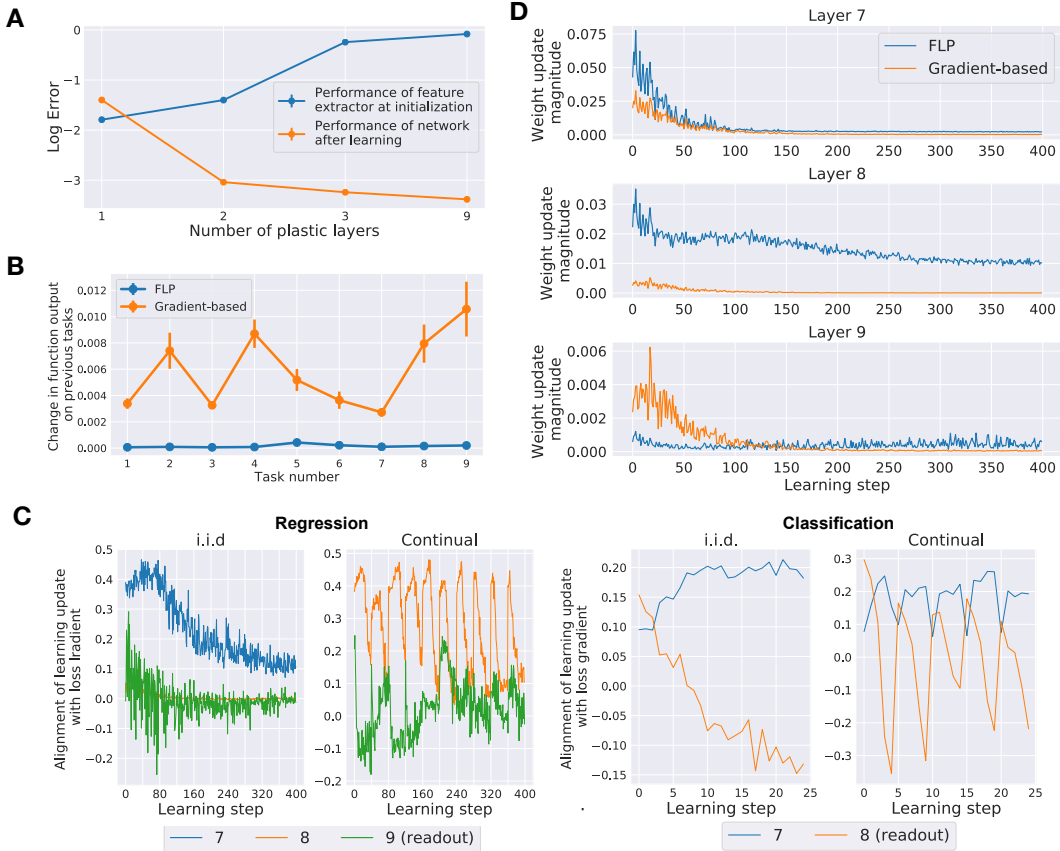

Figure 2: (A) In orange: performance of example networks with different numbers of plastic layers. In blue: performance of those same networks with non-readout weights frozen at their initializations, and readout weights trained for many epochs on i.i.d data within each lifetime. (B) Illustration of interference on the continual learning regression task. X-axis: number of tasks (of the ten total) that have been encountered thus far. Y-axis: The change in the network's outputs on data from previous tasks, comparing the network before and after it encounters the current task. (C) Correlation (cosine similarity) of the updates performed by the FLP networks with the negative gradient direction. (D) Magnitude of weight updates in different network layers over the course of learning the i.i.d. regression task, for the FLP and gradient-based networks.

Surprisingly, we found that the feedforward weights of FLP networks with many plastic layers, which ultimately learn to perform the task more effectively, are less effective at initialization than similar networks with fewer plastic layers. We quantified this phenomenon by freezing the initial weights of each network and fitting a linear readout to perform the tasks in the given task family – see Figure 2A.

This result alone does not preclude the possibility that a network could begin with an effective feature extractor and still learn the task well – the meta-optimization may have simply not chosen such a solution. To address this possibility, we took a network that had been meta-optimized in the feature reuse regime. Then we enabled feedback and plasticity in upstream layers while freezing the feedforward weight initialization, and continued meta-optimizing. This procedure failed to improve the performance of the network beyond the performance of a feature reuse network. Hence, initial weights that are optimized for feature reuse are distinct from those that enable learning in FLP networks. The results are suggestive of a tradeoff between adaptable networks – those that learn well from data – and networks that are task-ready (up to a linear readout) "from birth."

## 7.2 FLP networks mitigate cross-task interference and forgetting

We investigated the source of FLP's advantage on continual learning tasks. We hypothesized that FLP is able to mitigate "forgetting" – interference of new learning with previous knowledge. We quantified

this phenomenon in the regression task by measuring the average squared change – before vs. after encountering the data from one sinusoid – of the network's output on examples from earlier sinusoids. As shown in Figure 2B, the FLP network performs learning updates that have a drastically smaller effect on its previously learned behavior than the gradient-based baseline. The inner-loop learning performance trajectories (see Appendix C) are consistent with this observation – the gradient-based baselines match or exceed FLP networks' initial learning speed, but lose ground as more and more interfering tasks are encountered.

### 7.3 FLP networks perform weight updates that differ substantially from the loss gradient

The structure of FLP networks (with direct, fixed feedback pathways) prevents them from implementing gradient descent exactly, and the continual learning results above suggest that their learning strategy differs substantially from gradient descent. Once meta-trained, how distinct are their updates from those of gradient descent? We quantified this difference by examining the correlation between weight updates in FLP networks and updates that *would* be computed by gradient descent.

We found that the alignment of weight updates with the gradient direction was often weak, but also exhibited substantial diversity across layers (Figure 2C). Some network layers performed updates *negatively* correlated with the gradient update. Many layers exhibited periodicity in their gradient correlation corresponding to the structure of the continual learning task. Unlike feedback alignment, for which this correlation would consistently increase over training, FLP networks exhibit diverse trends over training steps. We also observed that, on the regression task, FLP networks perform larger weight updates in non-readout layers and smaller weight updates in the readout layer, relative to the gradient-based baseline (Figure 2D). Together, these phenomena indicate that the meta-learned feedback network learns in a manner that is qualitatively different from gradient-based learners.

## 8 Discussion

This work demonstrates that meta-learning procedures can optimize neural networks that learn online using feedback connections and local plasticity rules. These networks, once meta-optimized, use learning strategies that differ – sometimes in advantageous ways – from gradient-based optimization. Based on these results, we conjecture that there exists a space of learning algorithms, each with its own advantages and biases, that can be explored productively with meta-learning approaches. It is important to note that we have experimented with only a handful of benchmarks, each involving a rather narrow distribution of tasks. It is possible that FLP networks will exhibit different behavior on a wider array of learning problems. Nonetheless, we show that for standard benchmarks from the meta-learning literature, FLP is both competitive with and substantially different from backpropagation. Our results suggest several avenues for future work.

### 8.1 Extensions within the FLP framework

In this work, we focused on a simple, tractable instantiation of the FLP framework. There are many opportunities to branch out from the particular choices made here. For instance, meta-learning the plasticity rule itself, within some parameterized family [4, 22], could provide advantages over the simple Hebbian rule we adopted. Moreover, while we achieved surprisingly effective performance using feedback weights that are fixed within a lifetime, there is no *a priori* reason to enforce this constraint. One can imagine the feedback weights themselves being subject to plasticity along with the feedforward weights. There is also no reason feedback needs to take place only via direct pathways from the network output to its earlier layers. More complex feedback architectures may improve performance, and feedback from higher intermediate layers (with no target information) to lower layers could enable more effective forms of unsupervised learning.

### 8.2 Scaling the method

Meta-learning as implemented in this work is computationally expensive, as the meta-learner must backpropagate through the network's entire training procedure. In order to scale our approach, it will be important to find ways to meta-train networks that generalize to longer lifetimes than were used during meta-training, or to explore alternatives to backpropagation-based meta-training (e.g. evolutionary algorithms). The present work focused on the case of online learning from a manageable

amount of data, but the case of learning from prolonged exposure to large datasets is also of interest to neuroscientists and machine learning practitioners alike. Scaling the method substantially will be critical to exploring this regime.

Moreover, scaling to larger problems will enable a more precise characterization of the circumstances in which particular learning "ingredients" are necessary and useful. For instance, in preliminary experiments with our regression task, we found that enabling plasticity in all layers gives similar performance to enabling it in 3 layers. We suspect that the performance saturates as a function of the number of plastic layers because of the nature of these tasks, rather than a fundamental limitation of our algorithm, as we find the same trend holds for gradient-based meta-learners. Along the same lines, we found that enabling nonlinear feedback pathways also did not improve performance significantly. We suspect that different or more complex tasks might expose the value of even deeper and more sophisticated feedback pathways.

### 8.3 Biological realism

Our method avoids weight symmetry and nonlocality – two of the more unbiological aspects of gradient-based neural network training – but it still relies on some biologically unrealistic features. For instance, in the present implementation, the feedforward and feedback + update passes occur sequentially. A natural extension of our model would enable them to run in parallel, as in a recurrent network. This requires ensuring (through meta-learning, or perhaps a segregated dendrites model [9]) that feedforward and feedback signals do not interfere destructively. Moreover, our method requires the meta-learner to specify a precise feedforward and feedback weight initialization. Optimizing instead for a distribution of weight initializations or connectivity patterns might better reflect the limited precision with which connectivity can be specified by a genome [37]. Another direction is to apply meta-learning to understand particular biological learning systems (see [13] for an example of such an effort). Well-constrained, meta-optimized biological learning models might show emergence of learning circuits found in nature and suggest new ones to look for.

## Broader Impact

While the eventual impacts of our work are hard to predict because of its theoretical nature, we hope that it represents a step toward a better understanding of biological learning algorithms. Such an understanding may lead to more flexible artificial systems as well as advances in basic neuroscience. One concern is that these and related methods are computationally expensive, and their widespread adoption at scale could lead to significant energy consumption and/or raise barriers to entry in this field of research. It remains to be seen whether algorithmic advances can mitigate this issue.

## Acknowledgments and Disclosure of Funding

This work was supported by NSF NeuroNex Award DBI–1707398 and The Gatsby Foundation (GAT3708). JL is also supported by the DOE CSGF (DE–SC0020347). The authors declare no competing interests.

## Footnotes

[1]Source code for our experiments is available at github.com/jlindsey15/FeedbackAndLocalPlasticity

[2]We may take $a$ to be the pre or post-feedback presynaptic activations. The post-feedback case corresponds to a model in which neural activations are updated directly with feedback and Hebbian-style plasticity ensues. The pre-feedback case can be interpreted similarly, assuming a temporal eligibility trace for plasticity. Alternatively, the pre-feedback case could be interpreted as modeling an implementation in which feedback signals are propagated without affecting the neural activations used in feedforward computation. Possible biological implementations include a segregated dendrites model (see [9]), or feedback through neuromodulatory signals, with weight updates that are proportional to presynaptic and neuromodulatory activity (see [10]). We report results for the pre-feedback case; preliminary experiments suggest similar results are obtained in either case.

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
