[Supplementary Material]

# A  Appendix: Universality Proofs

We prove that sufficiently wide and deep neural networks with supervised feedback and local learning rules can approximate any learning algorithm. We borrow some of the notation and proof techniques from Finn et al. [1]. We suppose the network propagates an input $\mathbf{x}$ forward, receives a target signal $\mathbf{y}$ from a supervisor, propagates a function of $\mathbf{y}$ back to its neural activations (feedback), and undergoes synaptic plasticity according to a local learning role dependent on these activations. We let $\{(\mathbf{x}_k, \mathbf{y}_k)\}$ denote the training data, observed in that order, and $\mathbf{x}^\star$ denote the test input.

We want to construct a network architecture with feedforward function $\hat{f}(\cdot; \theta)$ and feedback function $g(\mathbf{y})$ such that $\hat{f}(\mathbf{x}^\star; \theta') \approx f_{\text{target}}(\{(\mathbf{x}, \mathbf{y})_k\}, \mathbf{x}^\star)$, where $\theta' = \theta_k$, $\theta_0 = \theta$, and $\theta_{k+1} = \theta_k + \Delta_{\theta_k}(\mathbf{y}, \hat{f}(\mathbf{x}; \theta_k))$. The update $\Delta_\theta(\mathbf{y}, \hat{f}(\mathbf{x}; \theta))$ is assumed to proceed according to a local learning rule that adjust a synaptic weight $w$ based on the previous weight value, the presynaptic activity $a$, and the postsynaptic activity $b$, where the values of $a$ and $b$ are taken following feedback propagation. We will consider Hebb's learning rule: $w \leftarrow w + \alpha(ab)$ and Oja's learning rule: $w \leftarrow w + \alpha(ab - b^2 w)$.

We let $\hat{f}$ be a deep neural network with $2N + 2$ layers and ReLU nonlinearities. We will ensure nonnegativity of the activations of the intermediate $2N$ layers, allowing us to treat them as linear. This simplification allows us to write the model as follows:

$$\hat{f}(\cdot; \theta) = f_{\text{out}}\left( \left( \prod_{i=1}^{N} \mathbf{W}_i^2 \mathbf{W}_i^1 \right) \phi(\cdot; \theta_{\text{ft}}); \theta_{\text{out}} \right),$$

where $\phi(\cdot; \theta_{\text{ft}})$ is an initial neural network with parameters $\theta_{\text{ft}}$. $\prod_{i=1}^{N} \mathbf{W}_i^2 \mathbf{W}_i^1$ is a product of $2N$ square linear weight matrices, and $f_{\text{out}}(\cdot; \theta_{\text{out}})$ is an output neural network with parameters $\theta_{\text{out}}$. We adopt corresponding notation of $\mathbf{B}_i^1, \mathbf{B}_i^2$ – feedback matrices projecting a function $\varphi(\mathbf{y})$ of the target (computed with a one-layer feedback network) to the outputs of the layers $\mathbf{W}_i^1, \mathbf{W}_i^2$ respectively, as well as $\beta_i^1, \beta_i^2$ (feedback strength) and $\alpha_i^1, \alpha_i^2$ (plasticity coefficients at $\mathbf{W}_i^1$ and $\mathbf{W}_i^2$). Concretely, the activation $x_i^j$ at the output of layer $\mathbf{W}_i^j$ is set to $\text{ReLU}((1 - \beta_i^j)x_i^j + \beta_i^j \mathbf{B}_i^j \varphi(\mathbf{y}))$, where $\beta_i^j \in [0, 1]$. We will ensure nonnegativity of the projection so that we may ignore the ReLU. The weights of layer $\mathbf{W}_i^j$ are then updated according to one of the following rules:

$$\mathbf{W}_i^j \leftarrow \mathbf{W}_i^j + \alpha_i^j x_i^j (\tilde{x}_i^j)^T \qquad \text{(Hebb's rule)}$$

$$\mathbf{W}_i^j \leftarrow \mathbf{W}_i^j + \alpha_i^j [x_i^j (\tilde{x}_i^j)^T - \text{diag}(x_i^j)^2 \mathbf{W}_i^j] \qquad \text{(Oja's rule)},$$

where $\tilde{x}_i^j$ refers to the activations at the layer preceding layer $x_i^j$, and $\text{diag}(x)$ denotes a square diagonal matrix with $x$ along the diagonal. We will conduct the proofs for Hebb's rule and Oja's rule in parallel, using $\mathcal{L}$ as an indicator variable – a value of 1 indicates we are using Oja's rule, and 0 corresponds to Hebb's rule. Hence we may write the learning rule compactly as follows:

$$\mathbf{W}_i^j \leftarrow \mathbf{W}_i^j + \alpha_i^j [x_i^j (\tilde{x}_i^j)^T - \mathcal{L} \cdot \text{diag}(x_i^j)^2 \mathbf{W}_i^j].$$

We set all $\mathbf{W}_i^2$ to be identity matrices, all $\beta_i^2$ to 0 (rendering the values of $B_i^2$ irrelevant), all $\beta_i^1$ to 1, all $\alpha_i^2$ to be 0, and all $\alpha_i^1$ to be a constant $\alpha$ (assumed in the rest of the proof to be sufficiently small). These choices specify an architecture consisting of feedforward layers organized in pairs. The first layer in each pair consists of a general feedforward matrix $\mathbf{W}_i^1$, which we will henceforth write simply as $\mathbf{W}_i$. The matrix $\mathbf{W}_i$ will undergo plasticity at rate $\alpha$ induced by the feedforward activations at its input and feedback-induced activations at its output from feedback matrix $B_i^1$ (which we will now write simply as $\mathbf{B}_i$). The second layer is a nonplastic identity transformation which effectively "shields" $\mathbf{W_{i-1}}$ from undergoing plasticity induced by the feedback projection $\mathbf{B}_i$. We assume no feedback propagation to and no plasticity in the feature extractor $\phi$ or output network $f_{\text{out}}$. Thus feedforward propagation is affected only by the $\mathbf{W}_i$, $\phi$, and $f_{\text{out}}$, and plasticity updates following feedback propagation will only modify the $\mathbf{W}_i$ matrices.

Now we expand $\hat{f}(\mathbf{x}^\star; \theta')$. We let $\mathbf{z}_k = \left( \prod_{i=1}^{N} \mathbf{W}_i \right) \phi(\mathbf{x}_k)$. After one step, each $\mathbf{W}_i$ is updated as follows:

$$\Delta_{\mathbf{W}_i} = \alpha \mathbf{B}_i \varphi(\mathbf{y}_1)\phi(\mathbf{x}_1)^T \left( \prod_{j=i+1}^N \mathbf{W_j} \right)^T - \alpha \mathcal{L} \cdot \mathrm{diag}(\mathbf{B}_i \varphi(\mathbf{y}_1))^2 \mathbf{W}_i.$$

and up to terms of $O(\alpha^2)$, the update is of the same form for all steps $k = 1, 2, ..., K$. We let $\alpha$ be small enough that higher-order terms in $\alpha$ can be ignored. Now

$$\Delta_{\mathbf{W}_i} = \sum_{k=1}^K \left[ \alpha \mathbf{B}_i \varphi(\mathbf{y}_k)\phi(\mathbf{x}_k)^T \left( \prod_{j=i+1}^N \mathbf{W_j} \right)^T - \alpha \mathcal{L} \cdot \mathrm{diag}(\mathbf{B}_i \varphi(\mathbf{y}_k))^2 \mathbf{W}_i \right] + O(\alpha^2).$$

Thus we can expand $\prod_{i=1}^N \mathbf{W}'_i = \prod_{i=1}^N (\mathbf{W}_i + \Delta_{\mathbf{W}_i})$ into the following form:

$$\prod_{i=1}^N \mathbf{W}_i + \alpha \sum_{k=1}^K \sum_{i=1}^N \left( \prod_{j=1}^{i-1} \mathbf{W_j} \right) \mathbf{B}_i \varphi(\mathbf{y}_k)\phi(\mathbf{x}_k)^T \left( \prod_{j=i+1}^N \mathbf{W_j} \right)^T \left( \prod_{j=i+1}^N \mathbf{W_j} \right) \tag{1}$$

$$- \alpha \mathcal{L} \sum_{k=1}^K \sum_{i=1}^N \left( \prod_{j=1}^{i-1} \mathbf{W_j} \right) \mathrm{diag}(\mathbf{B}_i \varphi(\mathbf{y}_k))^2 \left( \prod_{j=i}^N \mathbf{W_j} \right) + O(\alpha^2), \tag{2}$$

This expansion allows us to derive the form of $\mathbf{z}^\star$, the intermediate (pre-$f_{\mathrm{out}}$) output of the network acting on $\mathbf{x}^\star$:

$$\mathbf{z}^\star = \prod_{i=1}^N \mathbf{W}_i \phi(\mathbf{x}^\star) + \alpha \sum_{k=1}^K \sum_{i=1}^N \left( \prod_{j=1}^{i-1} \mathbf{W_j} \right) \mathbf{B}_i \varphi(\mathbf{y}_k)\phi(\mathbf{x}_k)^T \left( \prod_{j=i+1}^N \mathbf{W_j} \right)^T \left( \prod_{j=i+1}^N \mathbf{W_j} \right) \phi(\mathbf{x}^\star)$$
$$\tag{3}$$

$$- \alpha \mathcal{L} \sum_{k=1}^K \sum_{i=1}^N \left( \prod_{j=1}^{i-1} \mathbf{W_j} \right) \mathrm{diag}(\mathbf{B}_i \varphi(\mathbf{y}_k))^2 \left( \prod_{j=i}^N \mathbf{W_j} \right) \phi(\mathbf{x}^\star),$$

Note that appropriate choice of $\mathbf{W}_i$ and $\mathbf{B}_i$ allows us to simplify the form of $\mathbf{z}^\star$ in Equation 3 into the following:

$$\mathbf{z}^\star = \mathbf{G_0}\phi(\mathbf{x}^\star) + \alpha \sum_{k=1}^K \sum_{i=1}^N \mathbf{G_0}(\mathbf{G_{i-1}})^{-1} \mathbf{B}_i \varphi(\mathbf{y}_k)\phi(\mathbf{x}_k)^T \mathbf{G}_i^T \mathbf{G}_i^T \phi(\mathbf{x}^\star) \tag{4}$$

$$- \alpha \mathcal{L} \sum_{k=1}^K \sum_{i=1}^N \mathbf{G_0}(\mathbf{G_{i-1}})^{-1} [\mathrm{diag}(\mathbf{B}_i \varphi(\mathbf{y}_k))]^2 \mathbf{G_{i-1}}\phi(\mathbf{x}^\star) \tag{5}$$

where the $\mathbf{G}_i^T = \left( \prod_{i+1}^N \mathbf{W}_i \right)$ can be set to arbitrary invertible square matrices.

Now, our goal is to choose $\mathbf{G}_i^T$, $\mathbf{B}_i$, $\varphi$, and $\phi$ to ensure that the expression above contains a complete description of the values of $\{(\mathbf{x}, \mathbf{y})_k\}$ (up to permuting the order of the examples) and $\mathbf{x}^\star$. Since $f_{\mathrm{out}}$ can approximate any function to arbitrary precision, $\hat{f}(\mathbf{x}^\star; \theta') = f_{\mathrm{out}}(\mathbf{z}^\star)$ can approximate any function of $\{(\mathbf{x}, \mathbf{y})_k\}$ and $\mathbf{x}^\star$.

We set $\varphi(\mathbf{y}) = \mathrm{discr}(\mathbf{y})$, yielding a one-hot $d$-dimensional vector indicating the value of $\mathbf{y}$ up to arbitrary precision. We let $\phi$ (recall $\phi$ is a universal function approximator) have the following form:

$$\phi(\mathbf{x}) \approx \begin{bmatrix} 0 \\ \mathrm{discr}(\mathbf{x}) \\ \mathbf{0}_{J^2 d} \\ \mathrm{discr}(\mathbf{x}) \end{bmatrix},$$

where discr($\mathbf{x}$) is a one-hot $J$-dimensional vector indicating the value of $\mathbf{x}$ up to a discretization of arbitrary precision, and $\mathbf{0}_{J^2}$ is a zero vector of dimension $J^2$. Note that $\phi$ satisfies the requirement that all its outputs are nonnegative. We furthermore let $N = J^2$ and rewrite the layer index $i$ as a double index $(j, l)$ where $j$ and $l$ each range from 1 through $J$. For future reference let us denote the dimensionality of $\mathbf{y}$ as $d$. $\mathbf{G_{j,l}}$ and $\mathbf{B_{j,l}}$ are defined as follows:

$$
\mathbf{G_{j,l}} := \begin{bmatrix} 0 & \tilde{\mathbf{G}}_{j,l} & 0 & 0 \\ 0 & 0_{J \times J} & 0 & 0 \\ 0 & 0 & 0_{J^2 d \times J^2 d} & 0 \\ 0 & 0 & 0 & I_{J \times J} \end{bmatrix} + \epsilon I \quad \mathbf{B_{j,l}} := \begin{bmatrix} 0_{1 \times d} \\ 0_{J \times d} \\ \tilde{\mathbf{B}}_{j,l} \\ 0_{J \times d} \end{bmatrix} \tag{6}
$$

where $\tilde{\mathbf{G}}_{j,l}$ is a $1 \times J$ matrix containing ones in the $j$ and $l$ positions and zeroes elsewhere, the $\epsilon I$ is included to ensure the invertibility of $\mathbf{G_{j,l}}$, and $\tilde{\mathbf{B}}_{j,l}$ maps $\varphi(\mathbf{y})$ to a vector consisting of a stack of $J^2$ $d$-dimensional vectors, all of which are zero except the vector in the slot corresponding to $(j, l)$, which is $\varphi(\mathbf{y})$. That is,

$$
\tilde{\mathbf{B}}_{j,l}\varphi(\mathbf{y}) := \begin{bmatrix} \mathbf{0}_d \\ \vdots \\ \mathbf{0}_d \\ \text{discr}(\mathbf{y}) \\ \mathbf{0}_d \\ \vdots \\ \mathbf{0}_d \end{bmatrix} \tag{7}
$$

with $\varphi(\mathbf{y})$ appearing in the $J * j + l$ position.

Now we observe that:

$$
\phi(\mathbf{x})^T \mathbf{G_{jl}^T} \approx \begin{cases} \mathbf{e}_j^T & \text{if discr}(\mathbf{x}) \in \{\mathbf{e}_j, \mathbf{e}_l\} \\ \mathbf{0} & \text{otherwise} \end{cases} \quad \mathbf{G_{jl}}\phi(\mathbf{x}^\star) \approx \begin{cases} \mathbf{e}_j & \text{if discr}(\mathbf{x}^\star) \in \{\mathbf{e}_j, \mathbf{e}_l\} \\ \mathbf{0} & \text{otherwise} \end{cases}
$$

where the approximation in the equalities is due to the $\epsilon$ terms included to ensure invertibility.

As a result, we have:

$$
\mathbf{z}^\star \approx \mathbf{G_0}\phi(\mathbf{x}^\star) + \alpha \sum_{k=1}^{K} \begin{bmatrix} 0 \\ \mathbf{0}_J \\ \tilde{\mathbf{z}}_k^\star \\ \mathbf{0}_J \end{bmatrix},
$$

where $\tilde{\mathbf{z}}_k^\star \approx \begin{cases} v(\text{discr}(\mathbf{y}_k), \{j + J * l, l + J * j\}) & \text{if discr}(\mathbf{x}^\star) = \mathbf{e}_j \neq \mathbf{e}_l = \text{discr}(\mathbf{x}_k) \\ v(\text{discr}(\mathbf{y}_k), \{j + J * i | 1 \leq i \leq J\} \cup \{i + J * j | 1 \leq i \leq J\}) & \text{if discr}(\mathbf{x}^\star) = \mathbf{e}_j = \text{discr}(\mathbf{x}_k) \end{cases}$

with $v(\mathbf{a}, A)$ defined as the $J^2 d$-dimensional vector consisting of $J^2$ stacked $d$-dimensional vectors, all of which are zero except those located in the slots specified by the set $A$, which are set to $\mathbf{a}$.

Now we claim that $\{(\mathbf{x}, \mathbf{y})_k\}$ and $\mathbf{x}^\star$ can be decoded with arbitrary accuracy from $\mathbf{z}^\star$. Indeed, note that $\mathbf{G_0} = \prod_{i=1}^{N} \mathbf{W}_i$ contains an identity matrix in its last $J$-dimensional block, meaning that $\mathbf{G_0}\phi(\mathbf{x}^\star)$, and hence $\mathbf{z}^\star$, contains an unaltered copy of discr($\mathbf{x}^\star$) in its last $J$ dimensions, from which $\mathbf{x}^\star$ can be decoded to arbitrary accuracy. Given the value of $\mathbf{x}^\star$ we may also subtract $\mathbf{G_0}\phi(\mathbf{x}^\star)$ from $\mathbf{z}^\star$ and multiply by $\frac{1}{\alpha}$ to obtain an unaltered version of $\sum_{k=1}^{K} \tilde{\mathbf{z}}_k^\star$. Next, we may decode $\sum_{k=1}^{K} \tilde{\mathbf{z}}_k^\star$ in the following fashion. First, we can infer whether, and if so how many, of the $\mathbf{x}_k$ have the same discretization as $\mathbf{x}^\star$ by checking if any of the $J$ $d$-dimensional vectors in slot $j + J * j$ is nonzero, and if so, what its value is. If slot $j + J * j$ has nonzero value $\mathbf{c}$, we subtract $\mathbf{c}$ from all slots with index $j + J * i$ and $i + J * j$ for any $i$. Given discr($\mathbf{x}^\star$) $= \mathbf{e}_l$ the resulting vector, which we may call $\tilde{\mathbf{z}}_k^{\star\star}$, This leaves us with a vector which in each slot $j + J * l$ and $l + J * j$ indicates (by summing the $d$ components of the slot) how many times an $\mathbf{x}$ has been observed with discr($\mathbf{x}$) $= \mathbf{e}_j$ and (by looking at the nonzero components in the slot) counts of how many times every possible discr($\mathbf{y}$)

value was observed to correspond with that discr($\mathbf{x}$). Thus, the set $\{(\mathbf{x}, \mathbf{y})_k\}$ as well as $\mathbf{x}^\star$ can be decoded to arbitrary accuracy from $\mathbf{z}^\star$.

Since $f_{\text{out}}$ is a universal function approximator, we let $f_{\text{out}}(\mathbf{z}^\star)$ be the function that performs the decoding procedure above and then uses the inferred values of $\{(\mathbf{x}, \mathbf{y})_k\}$ and $\mathbf{x}^\star$ to approximate $f_{\text{target}}(\{(\mathbf{x}, \mathbf{y})_k\}, \mathbf{x}^\star)$ to arbitrary precision.

# B    Appendix: Experimental Details

## B.1    Hyperparameters

In Tables 1 and 2 we give values of hyperparameters used in our experiments. For most hyperparameters not essentially related to our algorithm, we inherited values from the published OML code. Through initial experimentation we determined that proper selection the meta-learning rate of the network plasticity coefficients was particularly important for performance. For every network, we performed a sweep for this hyperparameter over several orders of magnitude – the optimal value of each is indicated in the tables.. In the classification experiments, it was necessary to weight the plasticity in the penultimate and readout layers differently. Given our computational resource constraints, we first optimized over the penultimate layer plasticity learning rate and subsequently over the penultimate-readout plasticity ratio. We were also unable to achieve performance improvements by meta-optimizing over $\beta$ in our classification experiments, and so we clamped it at a value of 1.0. Our search was not exhaustive, and more experimentation could reveal a benefit of intermediate $\beta$ values for these tasks.

## B.2    Meta-training procedure

As noted in the text our training procedure for differed slightly from that of OML. On regression tasks, we found that a naive implementation of the gradient-based baseline (and the reported OML numbers) had difficulty exceeding the feature reuse regime – that is, plasticity in non-readout layers was not helpful. However, by initializing the gradient-based baseline with the feedforward weights uncovered by our FLP algorithm, we were able to substantially improve the performance of the gradient-based baseline. We used this initialization procedure so as to consider the strongest possible baseline – however, the difficulty of meta-optimizing the gradient-based learner may be of independent interest.

On classification tasks, we simplified the published OML training procedure so that the task used in meta-training (25 learning steps on 5 examples of 5 classes each) was the same as that being tested. In OML, a proxy task is used in which the network must learn one class during meta-training without forgetting classes it already has learned from other meta-training examples. Implemented naively, this procedure could lead to the network simply memorizing the meta-training classes without learning to learn novel classes. To mitigate this issue, the OML training procedure resets the readout weights corresponding to the current classes of interest at each meta-training step. This strategy enables OML to generalize well to longer sequences than those used during meta-training. However, applying this method to our framework is difficult, as resetting classifier weights disrupts the relationship between the meta-learned feedforward initialization and feedback weights. We hope to find ways around this issue in future work.

## B.3    Dataset details

On Omniglot, the meta-training dataset consists of the first 963 character classes, and the meta-testing dataset consists of the the remaining 660 classes. On mini-ImageNet, the first 64 classes and final 20 classes are used for meta-training and meta-testing, respectively.

## B.4    Evaluation details

Performance values for a single network were obtained by averaging results over 50 (for regression tasks) or 500 (for classification tasks) randomly sampled lifetims.

Table 1: Experimental parameters (regression)

| Parameter | Value |
|---|---|
| Meta LR (feedforward weights) | 1e-4 |
| Meta LR (feedback weights) | 1e-4 |
| Meta LR ($\beta$) | 1e-4 |
| Feedback strength(Initial $\beta$) | 0.5 |
| Meta LR (plasticity coefficients) | [1e-6, 1e-7, **1e-8**, 1e-9] |
| Initial plasticity | 0.0 |
| Minibatch size | 32 |
| Meta-training epochs | 20000 |
| Nonlinearity | ReLU |
| MLP layers | 9 |
| Layer width | 300 [1] |
| Meta-training optimizer | Adam |

Table 2: Experimental parameters (classification)

| Parameter | Value |
|---|---|
| Meta LR (feedforward weights) | 1e-4 |
| Meta LR (feedback weights) | 1e-4 |
| Feedback strength ($\beta$) | 1.0 (clamped) |
| Meta LR (plasticity coefficient, penultimate layer) | [1e-2, **1e-3** (omni), **1e-4** (img), 1e-5] |
| Meta LR (plasticity coefficient, readout-penultimate ratio) | [1e-5, **1e-4** (img), 1e-3, **1e-2** (omni), 1e-1] |
| Initial plasticity | 0.0 |
| Minibatch size | 1 |
| Meta-training epochs | 40000 |
| Nonlinearity | ReLU |
| Convolutional layers | 6 |
| Convolutional kernel size | 3 |
| Convolutional feature # | Omniglot: 128, Mini-ImageNet: 256 |
| Fully connected layers | 2 |
| Fully connected width | Omniglot: 128, Mini-ImageNet: 1000 |
| Meta-training optimizer | Adam |

## B.5 Runtime and computing infrastructure

Meta-training a network for 20,000 epochs takes roughly 3 days for the regression tasks on a single NVIDIA 1080 Ti GPU. Meta-training the classification networks takes roughly 1.5 days (for Omniglot) and 3 days (for mini-Imagenet) on a single NVIDIA 1080 Ti GPU.

# C   Appendix: Learning Trajectories

Figure 1: Inner-loop learning trajectories for the regression and Omniglot tasks in the i.i.d. and continual learning cases. These figures reflect performance on inner-loop training data and thus do not account for generalization error. In the continual case, performance is only assessed on the classes seen thus far. Error bars indicate standard errors.

## Footnotes

[1]Following the published code for OML ([2]), the sixth layer of the network has a width of 900.