[Reviews · NeurIPS 2020]

Review 1

Summary and Contributions: The authors use meta-learning on feedback connections and feedforward initializations and plasticity rates to overcome the non-biological constraints of locality and feedforward-feedback weight symmetry of backpropagation. Unlike hand-crafted solutions to this problem, their method performs as well or better than standard backprop on tested tasks. The weight updates after meta-learning seem to be qualitatively different from standard backprop.

Strengths: The work is complementary to the various directions already being pursued towards a biologically plausible version of backpropagation. The work suggests that other approaches than gradient descent might perform better and still be biologically plausible.

Weaknesses: Being a meta-learning approach on full weight matrices, it is not easy to compare this approach in a principled / equation-based way with backprop. While the authors do compare to an extent in section 7, this is rather empirical and ad hoc. Further, it is unclear if biology can encode such precise weight matrices genetically rather than distributions or plasticity rules, a point they concede in their lines 308-309. Personally, it would appear to me that meta-learning a parametrized plasticity rule on the feedback weights (and possibly also on the feedforward weights) would be more biological -- though they do leave this for future work. In lines 98-100, the authors state that "y, or the prediction error y − ŷ, is propagated through a set of feedback weights. We obtained better performance using prediction errors for the regression task, and using raw targets for the classification task". This should be explored further in section 7. What if both y and yhat are fedback?

Correctness: Seems reasonable.

Clarity: Overall the paper is well-written. However Section 7.1 needs to be made clearer.

Relation to Prior Work: The authors have compared their work reasonably fine with earlier work.

Reproducibility: Yes

Additional Feedback: Please address the points listed in weaknesses. [Update: In their rebuttal, the authors suggest that they plan to incorporate some of the issues mentioned in the weaknesses in a later paper. Even without this, the result that local learning rules could do better than backprop in some tasks is reasonably interesting, as typically local learning rules do not do as well. After going over the other reviewers' reviews and discussion, since the authors have agreed to incorporate R4's suggestions to references and nuanced contributions, I would keep my score.]


Review 2

Summary and Contributions: POST REBUTTAL The authors' rebuttal addresses my concerns. I am upholding my score on this paper. +++++++++++++ The authors propose a novel learning mechanism for feed-forward neural networks which rely on local plasticity rules for feed-forward connections and meta-learned feedback connections that provide error signals for these local plasticity changes. Building on established work that explores biologically plausible alternatives to back-propagation for credit assignment during network training. In this framework, feed-forward weights are update locally within the "lifetime" of the network, while the feedback weights and initialization weights of the feed forward connections are meta-learned in an outer loop. The authors perform simple numerical experiments that show their model can match SOTA meta-learning models, and even outperform them on a continual learning task. The authors provide a detailed analysis of learning dynamics that reveal their network employs a learning strategy that differs from that of gradient-based optimization, suggesting this mix of fast local plasticity rules and slow, meta-learned error feedback may have important advantages for neural network performance.

Strengths: The results presented in this paper are excellent. The methodology is clear, the analysis considers important alternative explanations to the claims, thus reinforcing the conclusions. The authors provide theoretical results that lay a good foundation for their experimental results. The discussion is clear and complements the results well. The results complement existing work in biologically plausible alternatives to back-propagation, yet they are novel in that they employ two "timescales" of learning, i.e. during the network's "lifetime" (inner loop) and across lifetimes for feedback weights "outer loop". This work is relevant for the NeurIPS community for two reasons: (1) it explores the important question of credit assignment during learning in the brain, and (2) it uncovers promising novel approaches for machine learning practitioners.

Weaknesses: A weakness of this work is the relative simplicity of the numerical experiments presented: one regression task, and two classification tasks. While there are understandable reasons for this —meta learning is computationally expensive and several variants of networks were trained for each task for analysis— the scope for purely ML practitioners is somewhat limited. Benchmark comparisons could also be extended to more models. A important detail that is not clear from the experiments (unless I missed it) is the difference in learning speed between gradient-based meta-learning baselines and the proposed approach. Do both methods converge as fast to the reported performances ? Another shortcoming with respect to biological plausibility is the following. While feed forward weights evolve w.r.t. local plasticity rules which are biologically plausible, feedback weights carrying error signals are still learned with back-propagation. This raises two problems: (1) there are marked differences between feedback and feed forward synaptic machinery, on the order of lifetime vs multi-lifetimes, which is difficult to reconcile with the current understanding of synaptic dynamics and anatomical changes in the brain. (2) The problem of "backpropagating" the error is only passed on the the outer loop, rather than being solved by a novel mechanism. While the authors address the limitations of their approach and point to future work to mitigate them, these two points should be explicitly addressed in the discussion.

Correctness: yes

Clarity: The paper is very well written.

Relation to Prior Work: yes

Reproducibility: Yes

Additional Feedback: Here are some suggestions to improve the paper: - Report and discuss training speed of proposed method compared to the gradient-based benchmark(s) - Reinforce the discussion about biological plausibility by addressing the above-mentioned weaknesses. Small suggestions: - Line 15 (Abstract): The sentence "Our results suggest the existence of a class of biologically plausible learning mechanisms that not only match gradient descent-based learning, but also overcome its limitations." is a little heavy-handed. I don't believe the method overcomes "all" of gradient descent limitations in continuous learning. Rephrasing would help. - Appendix references are broken - Line 110: "Note that βi = 0 corresponds to pure unsupervised Hebbian learning in layer i; thus, the βi parameter can be interpreted as interpolating between unsupervised and supervised learning. " Up to now, only activation updates were described but not learning updates. This note seems a little pre-emptive and was difficult to parse until weight updates were described later. - Line 130: a citation is needed to accompany the opening sentence of section 4.2 - A number of references are improperly reported. See e.g. [3], [24], [31].


Review 3

Summary and Contributions: This paper proposes a meta-learning algorithm that incorporates ideas from biologically-plausible learning algorithms. While the algorithm itself depends on backpropagation through multiple updates, making it highly non-plausible for biological systems, the idea is that this approach might learn biologically-plausible learning rules.

Strengths: The paper presents an interesting approach to finding new biologically-plausible learning algorithms.

Weaknesses: It was unclear to me whether this method is actually better than simple baselines, such as the baseline of training a single layer ("Feature Reuse" in the experiments). I suspect Feature Reuse does poorly in the 9-layer benchmark because the input information is washed out, and that running Feature Reuse on a shallow network would work better. Figure 2A might addresses this, but it was unclear what algorithm was being used. The experiments in Figure 1 only explore this approach for up to 2 deep layers. Presumably because the algorithm becomes unstable when used for more layers? This is not discussed, but this is the sort of thing I would expect to see.

Correctness: I think so, but I'm not certain.

Clarity: The paper is clear and well written.

Relation to Prior Work: The authors clearly discuss how their work relates to the related literature.

Reproducibility: Yes

Additional Feedback:


Review 4

Summary and Contributions: The paper proposes a meta-learning algorithm that uses gradient-free rules during learning and is meta-learned via gradient descent. These learning rules are meta-trained across simple regression and image classification datasets, the authors demonstrate the applicability of the rules for learning during meta-testing. The authors claim biological plausibility during meta-testing due to independent weights during the forward and backward pass and no necessity to store the intermediate states. The authors furthermore list the following contributions. 1. "We provide support for the idea that the credit assignment problem itself may be viewed as an optimization problem, amenable to solution via meta-learning." (l.79) 2. "Second, we provide evidence that faithfully approximating gradient signals is not the only route to effective credit assignment, and that for some problems there may be even more effective strategies" (l. 83) 3. "Third, we make preliminary attempts to understand the learning strategies uncovered by our meta-learning approach" (l.87) Not all of these contributions are novel as pointed out in the review.

Strengths: The authors investigate an important question: How can credit assignment be accomplished without gradients? This is important due to current limitations of backpropagation in terms of memory requirements and online learning. Additionally, the implausibility of backpropagation is of interest to the neuroscience community. The authors propose to use gradient based meta-learning to meta-learn learning rules that operate without backpropagation and the need of symmetric backward weights and retaining of the states during the forward pass. The introspection on the differences between the meta-learned updates and the gradient-descent updates is interesting.

Weaknesses: As pointed out before, one major advantage of their method is that during learning no backpropagation is required. The authors rightly mention that the same advantage applies to meta-learning in recurrent neural networks (e.g. [A], this particular work should also be cited) but do not compare to this method in their experiments. Not of all their contributions are novel: 1. "We provide support for the idea that the credit assignment problem itself may be viewed as an optimization problem, amenable to solution via meta-learning." (l.79). This has been shown many times before, e.g. [A,F,G] 2. "Second, we provide evidence that faithfully approximating gradient signals is not the only route to effective credit assignment, and that for some problems there may be even more effective strategies" (l. 83). This also has been shown many times before, e.g. [A,G,B] 3. "Third, we make preliminary attempts to understand the learning strategies uncovered by our meta-learning approach" (l.87). These experiments are indeed valuable. The main weakness of the present paper is that all experiments were only performed on extremely narrow data distributions (which is also pointed out in the conclusion). This is quite problematic because their meta-training procedure itself is not biologically plausible as it relies on gradient descent. Thus, if the meta-learned solution does not generalize, it is far inferior to gradient descent-based learning. Nevertheless, in the paper, they often claim that the found mechanism match or exceed gradient descent-based learning (e.g. l40, l214). Based on the experiments this cannot be concluded. While presumably intended to refer to the highly narrow data distributions used in the paper this is misleading and these claims should be toned down and clarified. Furthermore, only the top two to three layers of up to nine layers were adapted. This means credit assignment is very limited in the present experiments. The experiments in table 1 show that increasing the number of adapted layers does not always improve the final results. To summarize, the paper makes overly strong claims with regards to multiple aspects and not all contributions are as novel as they are presented. This has to be corrected to warrant acceptance. Despite of that, their specific implementation has sufficient novelty and should be of interest to the community.

Correctness: As pointed out above the claims for being comparable to backpropagation can not be made based on the empirical evaluation. Similarly, the claimed contributions need to be reformulated to be more specific to their presented implementation.

Clarity: Overall, the paper is well written but could be improved in several aspects: 1. The experimental setup (sec 5) is hard to follow and not entirely clear. We suggest a rewrite and adding a visualization of the different training phases and the data that is being used. 2. Section 6 introduced more about the experimental setup, mixed with results and analysis. We suggest moving all these specifications to section 5.

Relation to Prior Work: The authors do not mention foundational work on meta-learning. For example, they rightly discuss the connection to learning implemented in recurrent dynamics [31,32] but omit the original work in that area [A]. A connection to the literature of synaptic plasticity and learned learning rules is established but similar earlier work on fast weights [C, D] is missing. Differences to the synaptic learning rules [E] should be described more carefully, i.e. the missing feedback pathways. The authors write that their work supports "the idea that the credit assignment problem itself may be viewed as an optimization problem, amenable to solution via meta-learning." This was first done in [F,H,G,A]. [A] Hochreiter, S., Younger, A. S., & Conwell, P. R. (2001). Learning to learn using gradient descent. In International Conference on Artificial Neural Networks. [B] Ortega, P. A., Wang, J. X., Rowland, M., Genewein, T., Kurth-Nelson, Z., Pascanu, R., … Legg, S. (2019). Meta-learning of Sequential Strategies. [C] Schmidhuber, J. (1992). Learning to Control Fast-Weight Memories: An Alternative to Recurrent Nets. Neural Computation, 4(1), 131–139. [D] Schmidhuber, J. (1993). Reducing the ratio between learning complexity and number of time varying variables in fully recurrent nets. In International Conference on Artificial Neural Networks (pp. 460–463). (This is much like reference [2].) [E] Bengio, S., Bengio, Y., Cloutier, J., & Gecsei, J. (1992). On the optimization of a synaptic learning rule. In Preprints Conf. Optimality in Artificial and Biological Neural Networks (pp. 6–8). (Reference [4] in the NeurIPS submission.) [F] J. Schmidhuber. Evolutionary principles in self-referential learning, or on learning how to learn. Diploma thesis, TUM, 1987. [G] J. Schmidhuber. On learning how to learn learning strategies. TR FKI-198-94, TU Munich, 1994. [H] J. Schmidhuber. A self-referential weight matrix. Proc. ICANN'93, Amsterdam, pages 446-451. Springer, 1993. [2] J. Ba, G. E. Hinton, V. Mnih, J. Z. Leibo, and C. Ionescu. Using fast weights to attend to the recent past. In D. D. Lee, M. Sugiyama, U. V. Luxburg, I. Guyon, and R. Garnett, editors, Advances in Neural Information Processing Systems 29, pages 4331–4339. 2016. [31] J. X. Wang, Z. Kurth-Nelson, D. Tirumala, H. Soyer, J. Z. Leibo, R. Munos, C. Blundell, D. Kumaran, and M. Botvinick. Learning to reinforcement learn, 2016. [32] Jane X Wang, Zeb Kurth-Nelson, Dharshan Kumaran, Dhruva Tirumala, Hubert Soyer, Joel Z Leibo, Demis Hassabis, and Matthew Botvinick. Prefrontal cortex as a meta-reinforcement learning system. Nature neuroscience, 21(6):860–868, 2018.

Reproducibility: Yes

Additional Feedback: 1. You claim to use an online setup both for the iid and continual case. Nevertheless, in line 157 you write that you make use of batches. In line 171 you write that one example is presented at a time. Can you please clarify how you use batches? 2. Why was the feedback only done with direct pathways? This should be discussed in the paper. Edit after rebuttal: We read the rebuttal and the other reviews but must emphasize again the flaws pointed out in ours. It seems that the other reviewers largely overestimate the novelty of this work from a meta-learning perspective. That said, the authors have agreed to be more nuanced in their specific claims in the final version. Re: "I believe the experiments and analysis outline an important component about biological realism in neural network optimization: that gradient descent may not be the mechanism by which learning happens in the brain" We don't see any reason why the same couldn't have been achieved through the meta-learning recurrent neural network of 2001 [A] or certain other approaches we mentioned in our review. The authors don't offer any more evidence than related work that their approach could be a replacement for backpropagation, due to the extremely narrow task distributions their approach was tested on. It is well known that meta-learning recurrent neural networks can also learn to learn, at least on narrow task distributions [A,B]. Re: "but most of this previous work to be cited did not have biological realism in mind" We don't see why meta-learning recurrent neural networks, e.g., [A], would be less biologically realistic than the proposed approach. However, that alone should not be a reason for rejection, because perhaps the particular model architecture of the authors is indeed of particular interest to the neuroscience community. Is it? Given the assurance by the authors in their rebuttal to tone down their claims and relate this to previous work, we would be okay with accepting this submission, but the chair should insist on the required changes. In particular, their original claim of being equal or better than backpropagation needs to be put into perspective, considering the generality of backpropagation and the specificity of their meta-learned learning rule.

[Author Response · NeurIPS 2020]

We are grateful for the reviewers' constructive feedback.

**Novelty & limitations:** Reviewer 4 (R4) correctly points out that there exists a rich literature on meta-learning learning
rules for neural networks, and we did not adequately cite this literature. Regarding the high-level claims we made that
credit assignment may be viewed as an optimization problem and that there may be effective alternatives to gradient
descent for the problems we consider, we agree that our work is not the first to provide support for these ideas and
rather complements the references the Reviewer mentions by proposing and analyzing a concrete instantiation of them.
We agree with the Reviewer that our "specific implementation has sufficient novelty and should be of interest to the
community" and that, going forward, more work is needed to compare it to/integrate it with alternatives. We will both
cite the suggested references and be more nuanced in our claims about our specific advances in a final version.

R4 also points out an important limitation of this work, and most modern meta-learning studies in general, namely that
the meta-learned learning strategies are tailored to rather narrow task distributions. We agree that scaling our method,
e.g. by meta-training on a large, diverse set of tasks, is a high priority for future work. As R2 suspects, computational
constraints make doing so a challenge – we think tackling this challenge is worthy of another paper. We believe that our
paper, which uses contemporary (if imperfect) benchmarks from the meta-learning literature, is worth disseminating to
the community and will facilitate progress toward these even more ambitious goals.

**Training & performance:** R3 and R4 bring up the fact that our reported results involve plasticity in only the last
several layers of the network. To answer R3's question, this is not because the meta-training becomes unstable when
more plasticity is allowed – for instance, in the regression task, we have found that enabling plasticity in all layers gives
similar performance to enabling it in 3 layers. We suspect that the performance saturates as a function of the number of
plastic layers because of the nature of the tasks, rather than a fundamental limitation of our algorithm, as we find the
same trend holds for gradient-based meta-learners. We suspect that different or more complex tasks might require even
deeper plasticity. We will include these results and discussion in the camera-ready version of the paper.

R3 asks whether the feature reuse baseline would be stronger if we used a shallower network. We have now checked on
the regression task, and the performance is much worse using a 3-layer network (1.35 MSE) and about the same using a
6-layer network (0.05 MSE) compared to the 9-layer network that we used (0.05 MSE). Thus, we think ours is indeed a
fair baseline – it is not suffering from the network depth.

R2 asked how the learning speed of our method compares to that of the gradient-based baseline. Both the outer-loop
and inner loop learning speed are comparable. For instance, on the i.i.d. Omniglot task, both methods take $\sim$20,000
epochs to reach 5% error and $\sim$30,000 epochs to reach 3% error. In the inner loop, FLP learns a bit more quickly in early
iterations. We will add learning and meta-learning trajectories to the final version of this paper.

R4 asked how we used batches. In classification experiments, no batches are used (this is what "one example at a time"
referred to). In the regression experiments, the inner loop consists of 400 size-32 batches (to allow more examples to be
used in the inner loop while maintaining computational tractability). Note that each example is only presented once.
We will be more clear about these methods in a final version of the paper.

**Implementation choices:** Several Reviewers touched on assumptions concerning the nature of the feedback, including
fixed (within-lifetime) rather than plastic feedback weights and direct feedback from the readout layer. We made the
implementation decision of fixed feedback weights for simplicity, and to demonstrate how (remarkably) far one can get
with this approach. Going forward, plasticity of feedback weights is a natural extension, and may indeed be important
for scaling the method to harder and more diverse problems. Regarding the use of direct feedback from the output,
rather than from each hidden layer to the previous one, this was motivated by biological plausibility considerations,
as it remains an open problem whether and how error signals can be multiplexed with feedforward signals and also
transmitted backward to earlier network layers (though there are exciting proposals, e.g. Payeur et al. *bioRxiv* 2020).

R1 brought up the use of targets vs. errors for feedback. The idea of using both simultaneously is intriguing – we are
interested in exploring this, but it will take some time. We should note that our implementation decisions – targets for
classification, errors for regression – yield better performance both for FLP and the gradient-based baseline, so this
does not appear to be a quirk of our method. We will elaborate on this in the analysis section, as suggested.

R1 mentioned the difficulty in comparing our method to backpropagation, given that the outer loop is able to specify
weight initializations. We want to emphasize that our gradient-based baseline also used a meta-learned initialization
for fair comparison. We agree that full specification of the initialization is likely unbiological (at least for sufficiently
complex animals), and as mentioned in the Discussion, we hope to address this issue in future work.

R2 pointed out that backpropagation is used to learn the feedback weights in the outer loop. This is true, and we do not
attempt to model this outer loop optimization in a biological fashion. We consider the outer loop to be roughly analogous
to the biological processes of evolution and development and thus not subject to inner-loop locality constraints. That
said, we hope that follow-up work will model this outer-loop learning in a more biologically plausible fashion.

[Meta-Review · NeurIPS 2020]

The reviewers seem to agree that there is value in proposed work. After a discussion, based on the rebuttal, the consensus is that given that the authors integrate in the camera ready the details of the rebuttal (particularly the comments of R4) and *toning down* or being more precise in the claims being made, I think this work would be very interesting and useful to the community. Please do take into account this advice, as it will help the work to have maximal impact in the community and to not be misinterpreted or its claims to be abused.